# Pooling samples to increase testing capacity with Xpert Xpress SARS-CoV-2 during the Covid-19 pandemic in Lao People's Democratic Republic

**Vibol Iem**[1,2], **Phonepadith Xangsayarath**[3], **Phonenaly Chittamany**[1], **Sakhone Suthepmany**[1], **Souvimone Siphanthong**[1], **Phimpha Paboriboune**[4], **Silaphet Somphavong**[5], **Kontogianni Konstantina**[2], **Jahangir A. M. Khan**[6], **Thomas Edwards**[2], **Tom Wingfield**[2,7,8], **Jacob Creswell**[9], **Jose Dominguez**[10‡], **Luis E. Cuevas**[2‡]*

1 National Tuberculosis Control Center, Vientiane, Lao People's Democratic Republic, 2 Liverpool School of Tropical Medicine, Liverpool, United Kingdom, 3 National Center for Laboratory and Epidemiology, Vientiane, Lao People's Democratic Republic, 4 Center of Infectiology Lao Christophe Mérieux, Vientiane, Lao People's Democratic Republic, 5 Lao Association for Medical Laboratory Sciences, Vientiane, Lao People's Democratic Republic, 6 University of Gothenburg, Health Economics and Policy Unit, School of Public Health and Community Medicine, Gothenburg, Sweden, 7 Department of Global Public Health, WHO Collaborating Centre for Tuberculosis and Social Medicine, Karolinska Institutet, Solna, Sweden, 8 Tropical and Infectious Diseases Unit, Liverpool University Hospital NHS Foundation Trust, Liverpool, United Kingdom, 9 Stop TB Partnership, Innovations and Grants, Geneva, Switzerland, 10 Institut d'Investigació Germans Trias i Pujol, CIBER Enfermedades Respiratorias, and Universitat Autònoma de Barcelona, Spain

‡ These authors are joint senior authors on this work.
* Luis.Cuevas@lstmed.ac.uk

**Data Availability Statement:** All relevant data are within the paper and its Supporting Information files.

## Abstract

The COVID-19 pandemic created the need for large-scale testing of populations. However, most laboratories do not have sufficient testing capacity for mass screening. We evaluated pooled testing of samples, as a strategy to increase testing capacity in Lao PDR. Samples of consecutive patients were tested in pools of four using the Xpert Xpress SARS CoV-2 assay. Positive pools were confirmed by individual testing, and we describe the performance of the test and savings achieved. We also diluted selected positive samples to describe its effect on the assays CT values. 1,568 patients were tested in 392 pools of four. 361 (92.1%) pools were negative and 31 (7.9%) positive. 29/31 (93.5% (95%CI 77–99%) positive pools were confirmed by individual testing of the samples but, in 2/31 (6.5%) the four individual samples were negative, suggesting contamination. Pools with only one positive sample had higher CT values (lower RNA concentrations) than the respective individual samples, indicating a dilution effect, which suggested an increased risk of false negative results with dilutions >1:10. However, this risk may be low if the prevalence of infection is high, when pools are more likely to contain more than one positive sample. Pooling saved 67% of cartridges and substantially increased testing capacity. Pooling samples increased SARS-CoV-2 testing capacity and resulted in considerable cartridge savings. Given the need for high-volume testing, countries may consider implementation of pooling for SARS-CoV-2 screening.

**Funding:** This research was funded in part by the Global Fund to Fight AIDS, Tuberculosis and Malaria (LAO-T-GFMOH country grant), the Global Fund Covid-19 Response Mechanism (C19RM), the National Center for Laboratory and Epidemiology and the National Tuberculosis Control Center from the Ministry of Health of Lao PDR. TW is supported by grants from: the Wellcome Trust, UK (209075/Z/17/Z); the Medical Research Council, Department for International Development, and Wellcome Trust, UK (Joint Global Health Trials, MR/V004832/1), the Medical Research Council, UK (MR/V028618/1); the Academy of Medical Sciences, UK; and the Swedish Health Research Council, Sweden. The funders had no role in study design, data collection and analysis, decision to publish, or preparation of the manuscript.

**Competing interests:** The authors have declared that no competing interests exist.

## Introduction

The world is facing an unprecedented health crisis since the emergence of the Severe Acute Respiratory Syndrome Coronavirus 2 (SARS-CoV-2) resulting in the coronavirus disease-19 (Covid-19) pandemic [1,2]. Ministries of health have implemented unparalleled non-pharmacological (NPIs) and pharmacological interventions, with stay-at-home, curfews, masking, quarantine orders, and increasingly, new and repositioned treatments and immunizations. Since early in the pandemic, identifying infected individuals has been considered a key pillar to prevent onward transmission and to monitor the efficacy of NPIs. For this, testing is needed at a large scale. Covid-19 confirmation is based on the detection of SARS-CoV-2 RNA by nucleic acid amplification assays, such as real-time reverse-transcription polymerase chain reaction (RT-PCR) [3]. RT-PCR is highly sensitive and specific compared to rapid antigen testing [4], but the large number of tests required has generated test stockouts, delayed reporting and unmanageable workloads, outstripping the capacity of the laboratories [5–8].

Covid-19 was first reported in Lao People's Democratic Republic (PDR) in March 2020. Initial epidemic waves were controlled through NPIs but a large epidemic wave, which started in April 2021, resulted in the establishment of community transmission, leading to large numbers of test requests that exceeded the testing capacity of the country. In response, the National Covid-19 Task Force introduced pooled testing, to increase testing capacity and the efficiency of the diagnostic algorithm. Although ideally the method's performance should have been assessed before widespread implementation, the Task Force decided to implement the approach routinely, based on the urgent need to increase testing capacity.

In pooled testing, clinical samples of several patients are mixed (pooled) together and tested with a single test. If the test is negative, all the samples included in the pool are considered negative, while if the test is positive, all the samples are re-tested individually to identify the infected specimens [9]. Depending on the positivity rate of the pooled tests, pooling uses an overall lower number of tests than individual testing, increasing testing capacity, lowering costs, and saving time. This method has been used in diagnostic laboratories and blood banks to screen for infections, such as hepatitis B [10], and tuberculosis [11] and is increasingly reported for SARS-CoV-2 screening [12].

Here, we describe the agreement of pooled and individual testing in clinical specimens using the GeneXpert (Cepheid, US) with Xpert Xpress SARS-CoV-2 assays (Xpress) [13], changes in the assays cycle threshold (CT) values and the cost and processing time to detect SARS-CoV-2 within the context of the epidemic.

## Materials and methods

We conducted a prospective cross-sectional study from the 26[th] of April 2021 to the 24[th] of May 2021 in Vientiane, Lao PDR. The study was conducted under the authority of the Ministry of Health National Center for Laboratory and Epidemiology, which was responsible at that time for the mass screening of the population in four large open-air sites in the capital. Participants were invited to participate if they had Covid-19 symptoms, close contact with individuals with confirmed Covid-19 less than 14 days prior to disease onset; a history of travel to/from other countries, or if they had a diagnosis of Severe Acute Respiratory Infections or confirmed Covid-19 before hospital discharge. Both nasopharyngeal and oropharyngeal swabs were collected from each participant and put together in a single viral transport media tube. There were 1,568 consecutive samples included in the study, corresponding to the number of cartridges available at the National Tuberculosis Reference Laboratory during the study period. Samples were tested for SARS-CoV-2 using the pooling method with the Xpress cartridge, as shown in Fig 1. The Xpert Xpress is specifically designed to amplify sequences of the envelope

(E) and the nucleocapsid (N2) of the virus to generate tests results. If one or more SARS-CoV-2 nucleic acid targets (E, N2) has a CT within the valid range, the test reports a positive result. Pools were created by pipetting equal amounts (200 μL) of the individual samples directly into a container with the virus transport medium (BD Universal Viral Transport System, catalogue number 220220), and mixed together into a single use 2 mL cryovial tube. The pooled fluid was homogenized by soft pipetting-expelling to reduce risks of aerosols, loaded into an Xpress cartridge, and tested following the manufacturer's instructions [13]. If the pool tested positive, the four samples in the pool were then tested individually. If the pool tested negative, all samples were considered negative and were not re-tested. Individual Xpert Xpress results were notified for patient management by the Emergency Operation Centre.

CT values of pooled and individual samples were available for positive pools, to describe changes in viral loads. In addition, we conducted a bench evaluation of five clinical samples with known CT values to describe the dilution effect of the samples on the overall CT values. Samples were purposely selected if they had CT values <20, 20–25, 25–30, 30–35 and >35 and were diluted 1:2, 1:4, 1:6, 1:8, 1:10, 1:15 and 1:20 before testing. For this bench evaluation, 200 μL from the individual positive samples with known CT values were diluted using multiples of 200 μL fresh virus transport medium to replicate the desired dilution of the pools. Each diluted sample was tested with Xpert Xpress cartridges using 300μL per sample.

## Statistical analysis

All samples received were included in the analysis. Categorical data were summarized using descriptive statistics, with 95% confidence intervals (95% CI). We tested the agreement between the pools with positive Xpress results and the corresponding individual samples and estimated the cost and number of cartridges required to test all specimens using pooled and individual testing. Xpert Xpress costs were estimated at USD 19.80 per cartridge, as listed at wambo.org prices. Chi-squared tests were used to test for statistically significant differences between proportions. Changes in the CT values of the assays were described using correlations between the CT values of the non-diluted and diluted samples.

Sample size was not formally estimated as we were limited by the expected number of participants, the capacity of staff to conduct additional testing to their routine activities and the number of spare cartridges available for research purposes.

The datasets used and/or analysed during the current study are available from the corresponding author on reasonable requests for guideline development and systematic reviews.

## Ethics statement

Need for ethical approval and informed consent were waived by the National Center for Laboratory and Epidemiology, Lao PDR Ministry of Health. The Center is the delegated authority for Covid-19 testing. Permission was granted through the Emergency Operations Centre under Lao PDR Task force for Covid-19 Prevention, Control and Response.

## Patient and public involvement

It was not appropriate or possible to involve patients or the public in the design, or conduct, or reporting, or dissemination plans of our research.

## Results

The study included 898 (57.3%) males and 670 (42.7%) females. Young participants (under 35 years old) provided the majority of samples (1036, 66.1%), followed by 35–54-year-olds (446,

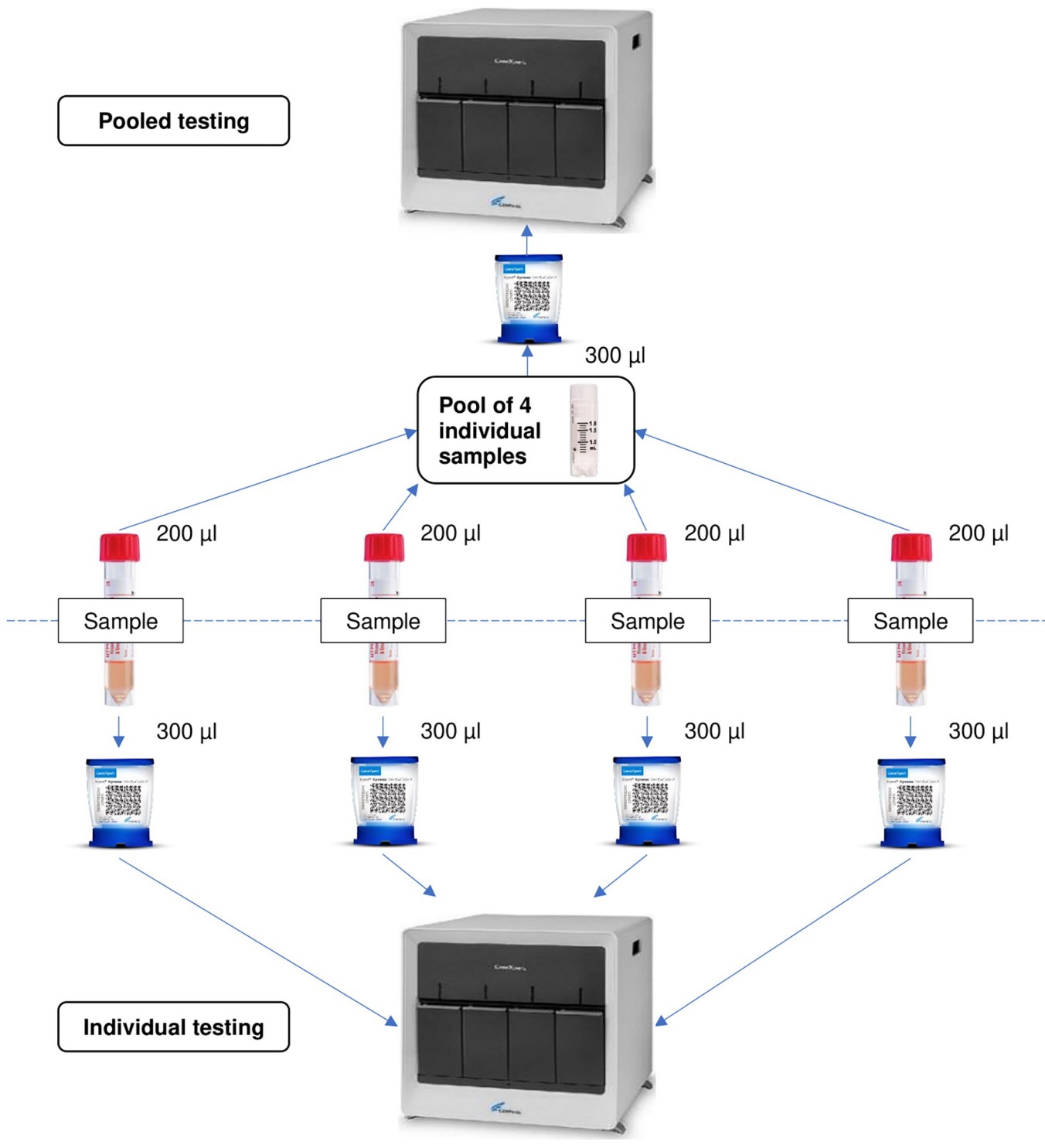

**Fig 1. Flow diagram of the sample processing.**

28.4%), with only a few samples belonging to participants $\geq$ 55 years (86, 5.5%), as shown in Table 1. The 1,568 samples were distributed into 392 pools, each pool containing four individual samples. Three hundred and sixty-one (92.1%) pools tested Xpress-negative and 31 (7.9%)

**Table 1. Baseline characteristics of participants with single and pooled Xpert Xpress SARS-CoV-2 results.**

| | | Xpert Xpress SARS-CoV-2 | |
|---|---|---|---|
| | | **Individual**<br>**n (%)** | **Pools**<br>**n (%)** |
| **Sex** | | **1,568** | NA |
| | Male | 898 (57.3%) | NA |
| | Female | 670 (42.7%) | NA |
| **Age** | | **1,568** | |
| | Mean (sd) (range) | 31.4 (12.6) (1–96) | NA |
| | <35 | 1036 (66.1%) | NA |
| | 35–54 | 446 (28.4%) | NA |
| | > = 55 | 86 (5.5%) | NA |
| **Individual Xpert Xpress SARS-CoV-2** | | **1,568** | **392** |
| Negative | | 1526 (97.3%) | 361 (92.1%) |
| Positive | | 42 (2.7%) | 31 (7.9%) |
| | <35 | 32 (76.2%) | NA |
| | 35–54 | 10 (23.8%) | NA |
| | > = 55 | 0 (0.0%) | NA |
| **SARS-CoV-2 positive** | | **42** | |
| | Male | 18 (42.9%) | NA |
| | Female | 24 (57.1%) | NA |
| **Number of positive samples included in Xpress-positive pools** | | | **31** |
| | 0 | - | 2 (6.5%) |
| | 1 | - | 20 (64.5%) |
| | 2 | - | 6 (19.4%) |
| | 3 | - | 2 (6.5%) |
| | 4 | - | 1 (3.2%) |

Xpress-positive. The samples of the 31 Xpress-positive pools were tested individually. Twenty (64.5%) of them contained only one positive sample, six (19.3%) contained two, two (6.4%) contained three and one (3.2%) contained four positive samples, for a total of 42 Xpert SARS-CoV-2 positive samples. Two (6.4%) positive pools did not contain positive samples when tested individually and were further re-tested for a different gene target using a different RT-PCR assay (Novel Coronavirus nucleic acid diagnostic kit PCR fluorescence probing (Sansure, China), on the CFX platform (BioRad, US)). However, all eight samples were still negative by this further assay [14]. Therefore 29 of the 31 positive pools were confirmed by individual testing, with an agreement of 94% (95%CI 77–99%).

The proportion of positive tests was similar for males and females (18/898, 2%, 95%CI 1.2–3.2% versus 24/670, 3.6%, 95%CI 2.4–5.4%, respectively, p = 0.06) and among adults < 35 years and 34–54 years old (32/1036, 3%, 95%CI 2.2–4.4% versus 10/446, 2.2%, 95%CI 1.1–4.2%, respectively, p = 0.4). However, none of the adults ≥ 55 years old was positive.

The pooled median CT for probe E of the pools with single positive samples was 21.7 (range 17.7–39.4) and 19.5 (range 14.8–35.4) for the individual samples (Table 2). The pooled CT values for probe E were higher than the individual values by a median of 2.3 (range 0.9–6.1, p = 0.2), as shown in S1 Table, S1 Fig. The median pooled CT values for probe N2 were 23.4 (range 19.5–43.6) and 21.2 (range 17.1–37.4) for the individual samples, respectively. The pooled CT value for probe N2 was higher than the individual CT values by a median of 2.2 (range 0.7–8.7, p = 0.2), (S1 Table, S1 Fig). The CT values of the two pools that tested positive

**Table 2. Median CT values of individual and pooled Xpert Xpress SARS-CoV-2 probe results.**

| Xpert Xpress SARS-CoV-2 CT values | | | | | |
|---|---|---|---|---|---|
| | Individual results n = 42 | | | Pooled results n = 29 | |
| Number of positive samples in the pool | CT Median | Min-Max | CT | Median | Min-Max |
| ΔCT Median | | | | | |
| 1 positive (n = 20) | | | | | |
| Probe E | 19.5 | 14.8–35.4 | 21.7 | 17.7–39.4 | 2.3 |
| Probe N2 | 21.2 | 17.1–37.4 | 23.4 | 19.5–43.6 | 2.2 |
| 2 positive (n = 6) | | | | | |
| Probe E | 22.7 | 15.1–39.9 | 19.7 | 17.2–32.9 | -0.8 |
| Probe N2 | 24.2 | 17.2–41.7 | 20.9 | 19.2–35.2 | -0.8 |
| 3 positive (n = 2) | | | | | |
| Probe E | 33.3 | 30.1–37.1 | 32.3 | 31.7–33.0 | -0.9 |
| Probe N2 | 34.2 | 31.8–37.1 | 33.6 | 33.6–33.7 | -0.5 |
| 4 positive (n = 1) | | | | | |
| Probe E | 37.0 | 31.9–38.2 | NA | NA | NA |
| Probe N2 | 39.8 | 34.2–41.9 | NA | NA | NA |

in the pool, but negative in the individual samples were 0 and 43.1 for probe E and N2 and 0 and 44.8 for the first and second pool, respectively, indicating they had high CT values and were late calls, corresponding to low SARS-CoV-2 RNA loads.

The CT values for the nine pools containing more than one positive sample are shown in S1 Table and their paired combination are shown in Fig 2. Six of the pools that contained two positive samples had a median probe E CT of 19.7 (range 17.2–32.9) compared to 22.7 (range 15.1–39.9) for the 12 positive individual samples within the pools and a median CT difference of -0.8 (range -9.6–0, p = 0.5). Similarly, the median pooled probe N2 CT was 20.9 (range 19.2–35.2), compared to 24.2 (range 17.2–41.7) for the individual samples, and a median difference of -0.8 (range -10–0.5, p = 0.5). A similar pattern was observed for the pool containing three and four positive samples, with median CT being higher in the pools than the individual samples (Fig 2, S1 Table).

The changes in the CT values for probes E and N of the five positive samples subjected to serial dilutions are shown in S2 Fig and S2 Table. CT values followed an almost linear increase in CT values across all samples and an increasing number of samples becoming undetectable at 1:10 and 1:20 dilution.

Cost analysis (Table 3) indicated testing 1568 participants would have required 1568 cartridges at a cost of USD 31,046.40. The pooling method required 392 cartridges to test in pools of four and 124 cartridges to re-test individually the 31 positive pools, resulting in 516 cartridges at a cost of USD 10,217.00. This represents a savings of 1052 (67%) cartridges, equivalent to USD 20,829.60. Using these same estimates, testing 1000 consecutive patients would require 329 cartridges instead of 1000 at a cost of USD 6.51 per participant. In Lao PDR's context, where laboratories receive a fixed allocation of cartridges, 1,000 cartridges would allow testing 3,040 individuals, which significantly increased testing capacity.

## Discussion

SARS-CoV-2 testing is often limited by the number of tests available. Assay shortages are multifactorial, from the limited production capacity for new assays, delayed procurement, a global shortage of RNA extractions kits, insufficient number of RT-PCR platforms and limited staff

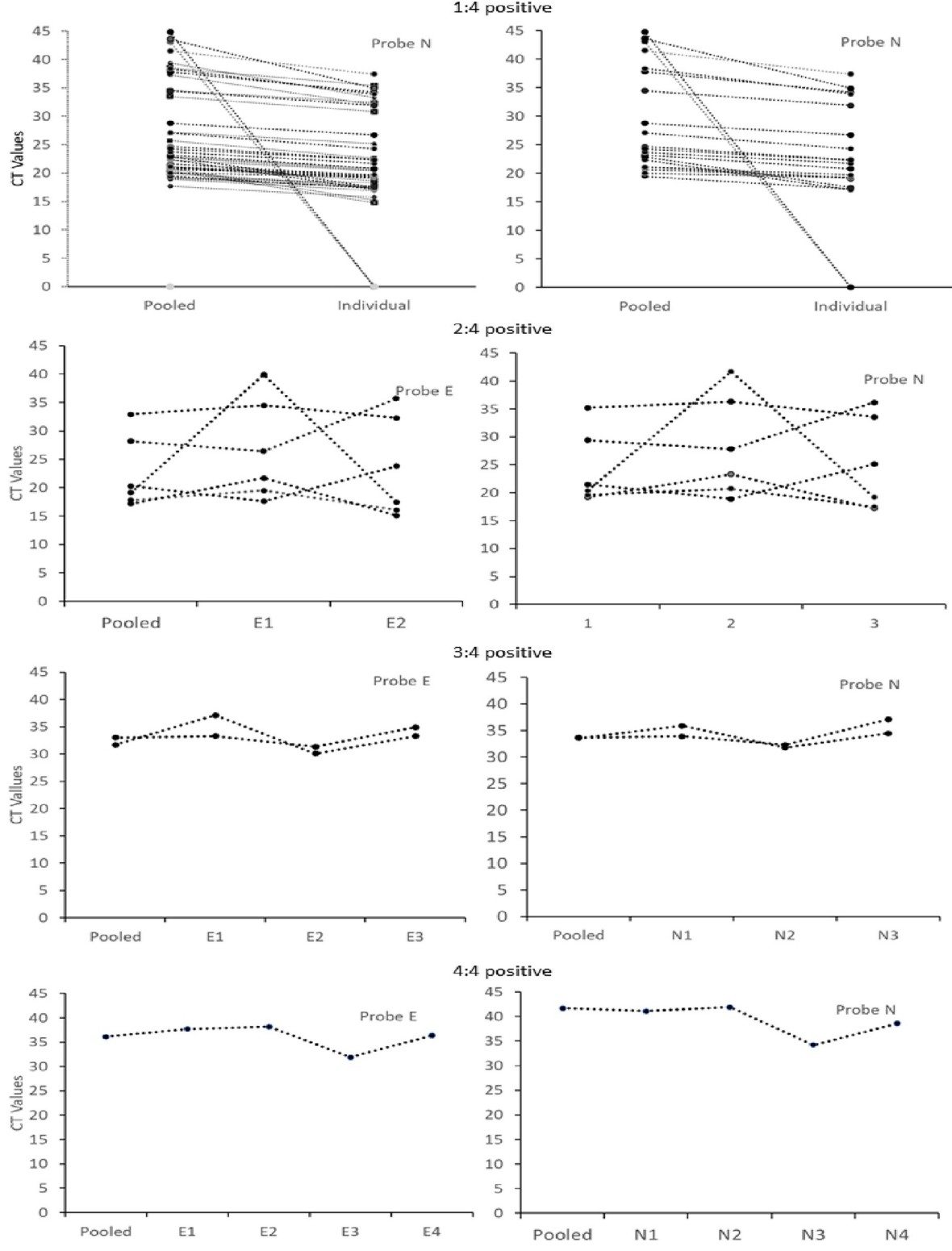

**Fig 2. CT values of samples containing 1, 2, 3 and 4 positive samples in a pool.**

**Table 3. Cost and diagnostic savings to screen 1000 consecutive patients using the pooling method and number of patients that could be tested with 1000 Xpert Xpress SARS-CoV-2 cartridges.**

| | | Pool of 4 | |
|---|---|---|---|
| | | **Individual testing** | **Pooled testing** |
| **Number of individuals tested** | | 1000 | 1000 |
| | Sensitivity | reference | NA |
| | Specificity | reference | NA |
| | Proportion positive | 2.7% | 7.9% |
| | Covid-19 cases confirmed | 27 | 27 |
| | Cartridges required | 1000 | 329 |
| | Cartridge costs (USD) | 19,800 | 6,514 |
| | Cartridge savings (USD) | 0 | **67.1%** |
| **Number tested with 1000 cartridges** | | | |
| | Number tested | 1000 | 3,040 |
| | Cartridge cost per patient (USD) | 19.80 | **6.51** |

for testing. It is thus unlikely that testing capacity will reach the number of test required in the short-term.

This study demonstrates that pooling samples can significantly increase testing capacity, while simultaneously reducing the resources needed for mass screening of SARS-CoV-2. The savings documented in our study, close to two thirds of the number of cartridges required for individual testing, are significant and were documented at a time when the proportion of pools testing positive was close to 8%. With the same resources required for individual testing, pooling allowed triplicating the number of people tested. Pooling has been reported to generate significant resource and time savings when screening for other infections, such as tuberculosis [11,15]. Savings are dependent upon the pool size and the proportion of pools that are positive. If the proportion of positive pools is low, e.g. at the nadir of an epidemic wave, most pools would be classified as negative and would not require further testing. However, if the proportion positive is high, many more pools would require individual testing [16]. Pooling therefore works well when there is a low prevalence of the pathogen, with more negative than positive results [17]. This is an important practical issue within the pandemic, as the proportion of positive pool varies rapidly during SARS-CoV-2 epidemic waves, with the introduction or removal of NPIs, the arrival of Variants of Concern, and mass gatherings [18].

All individual samples of two of the positive pools tested negative and remained negative when re-tested with a different assay. These pools had high CT values, suggesting they contained very low viral loads and were assumed to be false positive, caused by accidental cross contamination during sample preparation [19]. Although we had planned to test all individual samples in parallel to the pools, with the aim of exploring whether negative pools contained missed positive samples, this was not possible at the time of the emergency, as supplies were limited, and the Task Force prioritized implementing the approach to increase testing capacity. Previous studies have shown that the RNA concentrations of the individual and pooled tests are correlated, with a dilution effect in the pooled sample due to the lower sample volume used from each patient. This dilution effect increases the possibility of false negative pools, when the RNA concentration is below the limit of detection [17], and our result confirm this trend, with pools with single positive samples having higher CT values than individual samples. PCR CT values, however, are predictable, in that with 100% efficiency, the fluorescence should double each cycle. Therefore, if half of the target RNA is present, the CT value would be one CT later, thus a 1:2 dilution would result in a CT value +1 the undiluted sample, and 1:4 dilution in a

CT value of +2. Our data on serial dilutions (S2 Fig) seem to have increased CT values slightly more than expected, although these values are within the margin of error, and even though the increase of 1:10 to 1:20 is steeper, this would be expected at high dilution ratios. Interestingly, the dilution effect was not homogeneous, as pools with multiple positive samples often had the same or lower CT value than individual samples, thus indicating that the combination of multiple positive samples in a pool increases the total amount of genetic material and compensates for the dilution effect. Studies by others on pooling for SARS-CoV-2 testing from nasopharyngeal swabs have reported reduced CT values for pooled samples containing a single positive, hypothesizing that the PCR efficiencies were increased by a "carrier-RNA" effect caused by increased total cellular RNA in the samples [9,20]. Furthermore, pooling samples can lead to improved PCR efficiency and sensitivity in the case of a single positive sample containing PCR inhibitors, which are then diluted by pooling. However, no failed internal control results occurred on any of the Xpert Xpress runs of our study.

Our findings indicate that, in the Lao PDR context, the pooling method can increase the testing capacity by a factor of 3.04 compared to individual testing, with significant public health implications in situations of tests shortages and high demand for laboratory testing. In addition to resources savings, the rapid turnaround time with pooled testing can have a significant impact in terms of quarantine and isolation policies. By rapidly identifying positive clusters, the health authorities can trigger lockdowns in areas with confirmed outbreaks and ease the restrictions where there is no active transmission [21].

Other studies have reported that pooling with Xpert Xpress SARS-CoV-2 assay is reliable and that the dilution effect of multiple testing has limited effect on the sensitivity of the test [16,22]. Here we assessed the impact of the dilution on the CT values by performing serial dilution on samples of known CT values. Our results indicate the likelihood of false negative samples increases with the increasing dilution ratios, and that diluting the sample >1:10 results in significant losses of sensitivity. With a 1:20 ratio dilution, there was an increase in CT values >10, and consequently individual samples with low viral load and high CT values were missed by falling under the limit of detection. However, if the proportion of positive samples is high, the risk of false negative results may be minimized by the increased likelihood of samples containing more than one positive specimen in the pool.

## Study limitations

In other studies focusing on the assessment of the sensitivity and specificity of the pooling method, all samples are tested individually, and the pooling is conducted as an operational research to determine whether it could result in the same number of positive and negative patients as individual testing [23,24]. This study took place at a time when there was a consortium allocation established by WHO for Xpress tests, with 1,000,000 tests available globally, and Lao PDR was entitled for only 10,000 cartridges. Consequently, laboratories in the country received a fixed allocation of cartridges based on the burden of the disease in their catchment area and testing capacity with the shared GeneXpert platform, to ensure essential services for TB and HIV were not diverted. Therefore, given the limited resources available for mass screening, pooling was used as the reference method, assuming its sensitivity and specificity was acceptable, and that pooling was warranted based on public health needs. Individual testing was thus only done for individual samples in positive pools and therefore, we did not assess the sensitivity and the specificity of the method. Consequently, we don't know if, among the negative pools, there were positive samples that were missed. Moreover, the country decided to apply pools of four samples without assessment of the optimal pool size. A prior epidemiology analysis to determine the proportion of positive samples by province and district for the

different population groups would have allowed identifying whether larger pool sizes could have been more efficient. The pooling method is not a one-size-fits-all approach and statistical calculations using different combinations of pool sizes and positivity rate would have maximized the testing capacity and optimized the resources savings [22].

Furthermore, the cost analysis and the savings presented in this study did not include all savings that were generated around the pooling method, such as staff time, electricity, consumables, laboratory maintenance, samples transportation, wastes management and life expectancy of the GeneXpert machines.

## Conclusion

The pooling of samples for SARS-CoV-2 testing can be a useful strategy for testing when health systems are overwhelmed. This method can be rapidly implemented given the limited need for additional staff or sophisticated infrastructure. In a time where countries are facing shortage in laboratory supplies, with daily number of samples collected exceeding testing capacity, the pooling method can facilitate the expansion of testing in resource limited settings and accelerate the implementation or ease of NPIs.

## Supporting information

**S1 Fig. Correlation of individual and pooled Xpert Xpress Sars-CoV2 (positive pools only include those with single individual Xpert Xpress SARS-CoV2-positive sample).**
(TIF)

**S2 Fig. Effect of serial dilution on CT values of positive samples.**
(TIFF)

**S1 Table. CT values for the probes E and N2 for both individual and pooled results.**
(DOCX)

**S2 Table. Effect of dilution on CT values of positive samples.**
(DOCX)

**S1 Data.**
(XLSX)

## Acknowledgments

The authors would like to thank the National Center for Laboratory and Epidemiology of Lao PDR for facilitating access to samples and cartridges to carry out the study in an accelerated timeline. We are grateful to the laboratory technicians of the National TB Reference Laboratory (NTRL) and Center of Infectiology Lao Christophe Mérieux (CILM) for the careful management of samples and the additional work on top of their busy schedule, especially during the Covid-19 crisis.

## Author Contributions

**Conceptualization:** Vibol Iem, Kontogianni Konstantina, Jahangir A. M. Khan, Tom Wingfield, Jacob Creswell, Jose Dominguez, Luis E. Cuevas.

**Data curation:** Vibol Iem, Luis E. Cuevas.

**Formal analysis:** Vibol Iem, Jahangir A. M. Khan, Thomas Edwards, Tom Wingfield, Jose Dominguez, Luis E. Cuevas.

**Funding acquisition:** Vibol Iem, Sakhone Suthepmany, Tom Wingfield, Jacob Creswell, Luis E. Cuevas.

**Investigation:** Vibol Iem, Phonepadith Xangsayarath, Phimpha Paboriboune, Jose Dominguez, Luis E. Cuevas.

**Methodology:** Vibol Iem, Phonepadith Xangsayarath, Souvimone Siphanthong, Phimpha Paboriboune, Silaphet Somphavong, Kontogianni Konstantina, Jahangir A. M. Khan, Jacob Creswell, Jose Dominguez, Luis E. Cuevas.

**Project administration:** Vibol Iem, Phonenaly Chittamany, Sakhone Suthepmany, Tom Wingfield, Luis E. Cuevas.

**Resources:** Vibol Iem, Phonepadith Xangsayarath, Phonenaly Chittamany, Sakhone Suthepmany, Souvimone Siphanthong, Phimpha Paboriboune, Tom Wingfield, Jacob Creswell, Luis E. Cuevas.

**Supervision:** Vibol Iem, Phonenaly Chittamany, Sakhone Suthepmany, Souvimone Siphanthong, Silaphet Somphavong, Jahangir A. M. Khan, Tom Wingfield, Jose Dominguez, Luis E. Cuevas.

**Validation:** Vibol Iem, Thomas Edwards, Tom Wingfield, Jacob Creswell, Luis E. Cuevas.

**Visualization:** Vibol Iem, Luis E. Cuevas.

**Writing – original draft:** Vibol Iem, Sakhone Suthepmany, Souvimone Siphanthong, Silaphet Somphavong, Kontogianni Konstantina, Jahangir A. M. Khan, Thomas Edwards, Tom Wingfield, Jose Dominguez, Luis E. Cuevas.

**Writing – review & editing:** Vibol Iem, Phonepadith Xangsayarath, Phonenaly Chittamany, Phimpha Paboriboune, Tom Wingfield, Jacob Creswell, Jose Dominguez, Luis E. Cuevas.

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
