## [Decision Letter · Decision Letter 0]

29 Aug 2022

PONE-D-22-15022Pooling samples to increase testing capacity with Xpert Xpress SARS-CoV-2 during the Covid-19 pandemic in Lao People’s Democratic RepublicPLOS ONE

Dear Dr. Iem,

Thank you for submitting your manuscript to PLOS ONE. After careful consideration, we feel that it has merit but does not fully meet PLOS ONE’s publication criteria as it currently stands. Therefore, we invite you to submit a revised version of the manuscript that addresses the points raised during the review process. The reviewers are pleased with the manuscript and recommend its acceptance. However, minor details are required regarding sample dilution etc. Kindly address these minor comments and revert.

We look forward to receiving your revised manuscript.

Kind regards,

Chika Kingsley Onwuamah, Ph.D.

Academic Editor

PLOS ONE

https://journals.plos.org/plosone/s/file?id=ba62/PLOSOne_formatting_sample_title_authors_affiliations.pdf".

“This research was funded in part by the Global Fund to Fight AIDS, Tuberculosis and Malaria (LAO-T-GFMOH country grant), the Global Fund Covid-19 Response Mechanism (C19RM), the National Center for Laboratory and Epidemiology and the National Tuberculosis Control Center from the Ministry of Health of Lao PDR.

TW is supported by grants from: the Wellcome Trust, UK (209075/Z/17/Z); the Medical Research Council, Department for International Development, and Wellcome Trust, UK (Joint Global Health Trials, MR/V004832/1), the Medical Research Council, UK (MR/V028618/1); the Academy of Medical Sciences, UK; and the Swedish Health Research Council, Sweden.”

4. Thank you for stating the following in the Funding Section of your manuscript:

“This research was funded in part by the Global Fund to Fight AIDS, Tuberculosis and Malaria (LAO-T-GFMOH country grant), the Global Fund Covid-19 Response Mechanism (C19RM), the National Center for Laboratory and Epidemiology and the National Tuberculosis Control Center from the Ministry of Health of Lao PDR. TW is supported by grants from: the Wellcome Trust, UK (209075/Z/17/Z); the Medical Research Council, Department for International Development, and Wellcome Trust, UK (Joint Global Health Trials, MR/V004832/1), the Medical Research Council, UK (MR/V028618/1); the Academy of Medical Sciences, UK; and the Swedish Health Research Council, Sweden. The sponsors had no role in the study design; collection, analysis and interpretation of data; the writing of the manuscript; the decision to submit the manuscript for publication.”

“This research was funded in part by the Global Fund to Fight AIDS, Tuberculosis and Malaria (LAO-T-GFMOH country grant), the Global Fund Covid-19 Response Mechanism (C19RM), the National Center for Laboratory and Epidemiology and the National Tuberculosis Control Center from the Ministry of Health of Lao PDR.

TW is supported by grants from: the Wellcome Trust, UK (209075/Z/17/Z); the Medical Research Council, Department for International Development, and Wellcome Trust, UK (Joint Global Health Trials, MR/V004832/1), the Medical Research Council, UK (MR/V028618/1); the Academy of Medical Sciences, UK; and the Swedish Health Research Council, Sweden.”

Please include your amended statements within your cover letter; we will change the online submission form on your behalf."""

“The authors have no conflicts of interest to declare.”

Reviewers' comments:

Reviewer's Responses to Questions

**Comments to the Author**

1. Is the manuscript technically sound, and do the data support the conclusions?

Reviewer #1: Yes

Reviewer #2: Yes

2. Has the statistical analysis been performed appropriately and rigorously? 

Reviewer #1: Yes

Reviewer #2: Yes

3. Have the authors made all data underlying the findings in their manuscript fully available?

Reviewer #1: Yes

Reviewer #2: Yes

4. Is the manuscript presented in an intelligible fashion and written in standard English?

Reviewer #1: Yes

Reviewer #2: Yes

5. Review Comments to the Author

Reviewer #1: This is a very interesting manuscript that will find much deserved relevance in developing world where shortage of diagnostics is common. The manuscript has been well written with clear justification and sound conclusion.

Reviewer #2: From the methodology provided, samples were collected, pooled and tested, there is a need to add information on what was used in diluting the samples. which of the samples were diluted, the entire pool or individual samples that makes up the pool?

Author should give a brief description of the analysis in the methodology to specifically show what Probe E and N2 signifies. And for clarity what gene was targeted.

May include catalogue number/make of single viral transport media tube

6. PLOS authors have the option to publish the peer review history of their article (what does this mean?). If published, this will include your full peer review and any attached files.

Reviewer #1: No

Reviewer #2: **Yes: **Francisca Obiageri Nwaokorie

---

## [Author Response · Author response to Decision Letter 0]

4 Sep 2022

Response to Reviewers

Edited as suggested.

In the manuscript line 150-153, it is stated: “Need for ethical approval and informed consent were waived by the National Center for Laboratory and Epidemiology, Lao PDR Ministry of Health. The Center is the delegated authority for Covid-19 testing. Permission was granted through the Emergency Operations Centre under Lao PDR Task force for Covid-19 Prevention, Control and Response.” Therefore, we haven’t changed the text. However, we have added the level 2 heading “Ethics statement” at the beginning of the paragraph.

The same statement is also already in the online submission form.

3. Thank you for stating the following financial disclosure. Please state what role the funders took in the study. If the funders had no role, please state: ""The funders had no role in study design, data collection and analysis, decision to publish, or preparation of the manuscript.""

We have removed the section line 450-452, that stated: “The sponsors had no role in the study design; collection, analysis and interpretation of data; the writing of the manuscript; the decision to submit the manuscript for publication”. 

We have now included the following statement “The funders had no role in study design, data collection and analysis, decision to publish, or preparation of the manuscript” in the revised cover letter as suggested.

4. Thank you for stating the following in the Funding Section of your manuscript.

Please remove any funding-related text from the manuscript… 

We have removed all funding-related text as suggested.

…. and let us know how you would like to update your Funding Statement. Currently, your Funding Statement reads as follows:

“This research was funded in part by the Global Fund to Fight AIDS, Tuberculosis and Malaria (LAO-T-GFMOH country grant), the Global Fund Covid-19 Response Mechanism (C19RM), the National Center for Laboratory and Epidemiology and the National Tuberculosis Control Center from the Ministry of Health of Lao PDR.

TW is supported by grants from: the Wellcome Trust, UK (209075/Z/17/Z); the Medical Research Council, Department for International Development, and Wellcome Trust, UK (Joint Global Health Trials, MR/V004832/1), the Medical Research Council, UK (MR/V028618/1); the Academy of Medical Sciences, UK; and the Swedish Health Research Council, Sweden.”

We confirm that this funding statement is correct.

We have edited the covert letter as suggested.

5. Thank you for stating the following in your Competing Interests section. This information should be included in your cover letter; we will change the online submission form on your behalf.

We have edited the cover letter as suggested.

We have removed the duplicate ethical statement line 425-429. The ethical statement now only appears once at line 150-153, in the Methods section, as suggested.

The reference list was built using endnote, and is accurate with no retracted references. 

Reviewer #2: From the methodology provided, samples were collected, pooled and tested, there is a need to add information on what was used in diluting the samples. which of the samples were diluted, the entire pool or individual samples that makes up the pool?

We have added the following text to clarify methodology at line 118 as suggested by the reviewer: “Pools were created by pipetting equal amounts (200 µL) of the individual samples directly into a container with the virus transport medium (BD Universal Viral Transport System, catalogue number 220220), and mixed together into a single use 2 mL cryovial tube.” 

Another statement was added at line 132: “For this bench evaluation, 200 µL from the individual positive samples with known CT values were diluted using multiples of 200 µL fresh virus transport medium to replicate the desired dilution of the pools”.

Author should give a brief description of the analysis in the methodology to specifically show what Probe E and N2 signifies. And for clarity what gene was targeted.

Probe E and N2 refer to probes targeting genes E and N2 respectively. A statement was added at line 118 to clarify the terminology as suggested: “The Xpert Xpress is specifically designed to amplify sequences of the envelope (E) and the nucleocapsid (N2) of the virus to generate tests results. If one or more SARS-CoV-2 nucleic acid targets (E, N2) has a CT within the valid range, the test reports a positive result.”

May include catalogue number/make of single viral transport media tube

The reference was added at line 119 as suggested.

---

## [Editor Report · Decision Letter 1]

13 Sep 2022

Pooling samples to increase testing capacity with Xpert Xpress SARS-CoV-2 during the Covid-19 pandemic in Lao People’s Democratic Republic

PONE-D-22-15022R1

Dear Dr. Iem,

We’re pleased to inform you that your manuscript has been judged scientifically suitable for publication and will be formally accepted for publication once it meets all outstanding technical requirements.

Kind regards,

Chika Kingsley Onwuamah, Ph.D.

Academic Editor

PLOS ONE
---

## [Editor Report · Acceptance letter]

20 Sep 2022

PONE-D-22-15022R1 

Pooling samples to increase testing capacity with Xpert Xpress SARS-CoV-2 during the Covid-19 pandemic in Lao People’s Democratic Republic 

Dear Dr. Iem:

I'm pleased to inform you that your manuscript has been deemed suitable for publication in PLOS ONE. Congratulations! Your manuscript is now with our production department. 

Kind regards, 

on behalf of

Dr. Chika Kingsley Onwuamah 

Academic Editor

PLOS ONE